# Does Forest Industries in China Become Cleaner? A Prospective of Embodied Carbon Emission

Lanhui Wang [1], Zichan Cui [1], Jari Kuuluvainen [2] and Yongyu Sun [3,*]

1 School of Economics and Management, Beijing Forestry University, Beijing 100083, China; wanglh@bjfu.edu.cn (L.W.); zichan_cui@163.com (Z.C.)
2 Department of Forest, University of Helsinki, 00014 Helsinki, Finland; jari.kuuluvainen@helsinki.fi
3 Research Institute of Resource Insects, Chinese Academy of Forestry, Kunming 650224, China
* Correspondence: cafsdren@163.com

**Abstract:** Forests and the forest products industry contribute to climate change mitigation by sequestering carbon from the atmosphere and storing it in biomass, and by fabricating products that substitute other, more greenhouse-gas-emission-intensive materials and energy. This study investigates primary wood-working industries (panel, furniture, pulp and paper) in order to determine the development of carbon emissions in China during the last two decades. The input–output approach is used and the factors driving the changes in $CO_2$ emissions are analyzed by Index Decomposition Analysis–Log Mean Divisia Index (LMDI). The results show that carbon emissions in forest product industries have been declining during the last twenty years and that the driving factor of this change is the energy intensity of production and economic input, which have changed dramatically.

**Keywords:** embodied carbon; forest industry; energy intensity; energy structure





## 1. Introduction

Carbon emissions are a global concern for human sustainable development. According to the IEA (International Energy Agency), the global carbon emissions in 2019 were 33.6 Gt, an increase of 9.8% per cent over 2010, and emissions are expected to increase further to 36.4 Gt by 2030 (Data source: https://www.iea.org/articles/global-co2-emissions-in-2019 (accessed on 21 February 2021); https://www.iea.org/data-and-statistics?country=WORLD&fuel=CO2%20emissions&indicator=Total%20CO2%20emissions (accessed on 21 February 2021)). China has become the world's largest total $CO_2$ emitter and is facing enormous challenges in taking responsibility to reduce its emissions. Meanwhile, China was the largest importer and exporter forest products during 2008–2017, according to FAO statistics (FAO, FAO Forest Product Yearbook 2017). The value of the trade of forest products increased from USD 6.27 billion in 2008 to USD 16.88 billion in 2017 (Data source: http://www.gov.cn/xinwen/2019-12/04/content_5458275.htm (accessed on 21 February 2021)). China has become the largest importer of logs, pulp and sawnwood and the largest exporter of wood furniture and wood-based panels. Forest industries (panel, furniture, pulp and paper industry) are sectors with high energy consumption and high carbon emissions in terms of per unit GDP contribution. Forest product industries accounted for 1.5% energy consumption, with only a 0.56% GDP contribution in 2015 in China (calculated based on the energy data released in the Energy Report of the State Bureau of Statistics). With expanding trade in forest products, both domestically and internationally, emissions of carbon by the forest industries (FI) are increasing and China is taking much of the blame for this increase. In order to reduce the emissions, it is important to understand the mechanisms behind carbon emissions in the forest sector.

In the path towards reducing carbon emissions, it is of interest to study whether the forest industries in China have been able to reduce their $CO_2$ emissions, as well as

which factors are important determinants of the development. This paper will investigate these questions.

### 1.1. Embodied Carbon Theory

The term "embodied carbon" refers to carbon dioxide emitted at all stages of a product's manufacturing process, from the mining of raw materials through the distribution process, to the final product provided to the consumer. The concept of embodied carbon can be traced back to 1974. Embodied energy was proposed by the Energy Analysis Group of the International Federation of Institutes for Advanced Studies in order to measure the energy consumed throughout a supply chain or life-cycle. Later on, this concept was expanded, developed and applied to carbon emissions (Odum, 1996 [1]; Allan, 1997 [2]).

### 1.2. Literature Review

There is by now extensive literature on embodied carbon emission measurement at all scales—globally, inter-regionally and nationally. Tian, Liao & Wang (2015) studied spatial–temporal variations in embodied carbon emissions in global trade flows [3]. Liu et al. (2016) argued that embodied $CO_2$ emissions in Chinese exports were overestimated by 20% at the national level, with huge differences at the sector level, for 2007 [4]. Particularly, the topic related to China and its international trade has gained much research interest (Wu, Geng, Dong, Fujita, & Tian, 2016 [5]; Z. Li et al., 2017 [6]; Y. Zhao, Wang, Zhang, Liu, & Ahmad, 2016 [7]; Qi Ye, 2008 [8]; Yin, Cheng Ming, 2010 [9]; Song, 2012 [10], Zhao & Liu, 2010 [11]). Jiang et al. (2015) investigated the factors driving the inter-regional carbon flows in China [12]. These studies investigate the dynamics, the increasing future tendencies and the total amount of the embodied carbon in the main industries or in all industries from the perspective of international or inter-regional trade. However, less light has been shed on the forest sector compared to other primary industries such as steel building or textiles. Q. Zhao, Ding, Wen & Toppinen (2019) studied the carbon footprint in the Chinese pulp and paper industry [13]. Bai Weirong (2013) discussed the carbon emissions of wood furniture and considered a carbon reduction strategy [14]. Tian et al. (2015) studied the energy flows and carbon footprint of the pulp paper industry [3]. Sun et al. (2017) carried out a meta-analysis on carbon emissions in the pulp and paper industry [15]. Elias et al. (2020) investigated the impact of structural changes in wood-using industries on net carbon emissions in Finland [16]. Forest industry produces diverse products such as panels, furniture, pulp and paper, while there are only a few studies on carbon emissions in the forest industry as a whole. Furthermore, the development of these emissions over time has not received much attention. The goal of this study is to investigate the carbon embodied emission dynamics in the forestry industry in China over the past two decades and reveal the determinants which drive the changes.

The organization of the paper is as follows: in the next section, measurement of the embodied carbon method is described. In Section 3, the calculation of China's forest industry is presented. Section 4 presents the analysis of the factors driving the carbon emission changes, and then Section 5 presents the conclusions and discussion.

## 2. Methods

### 2.1. Method to Calculate the Embodied Carbon

Embodied carbon can be calculated by either top-down or bottom-up methods. This paper will select an optimal and feasible method with consideration of data accessibility and objectives.

The top-down method uses input–output analysis and has often been applied to estimate embodied energy, $CO_2$ emissions, pollutants and land appropriation from international trade activities (Wyckoff and Roop, 1994 [17]; Schaeffer and deSá, 1996 [18]; Machado et al., 2001 [19]; Munksgaard and Pedersen, 2001 [20]; Muradian et al., 2002 [21]; Hubacek and Giljum, 2003 [22]; Shui and Harriss, 2006 [23]). The input–output method takes the national economy as an integrated entity, in order to study the quantitative

relations between its various sectors. Connections between the sectors can be analyzed as a whole, with each sector consuming products from other sectors, so the embodied carbon emissions can be measured by the connections and energy consumption between the various sectors by input–output analysis. This method can be used to analyze a country's embodied carbon in international trade as a whole, but it has difficulties at the sectoral level. Input–output tables are expressed in terms of value added by sector and each sector spans a number of different, specific products, each of which will have different carbon-to-value-added ratios, or carbon coefficients. Since the sector carbon coefficients are estimated averages of those ratios for all the products in each sector, they are not particularly useful for calculating the embodied carbon attributable to a given product.

In the bottom-up approach, life-cycle analysis (LCA, ISO 2006 [24]) can be used. This approach needs detailed records of raw material production for each final product in the whole production process, which means that a great deal of effort is needed in integrating various data sources, with some data being difficult to access and some even unavailable. Because forest product industries deal with various forms of processing products, it is difficult to use the life-cycle method to account for forest products in all categories. Moreover, LCA requires detailed data on the energy consumption of each processing product, which are difficult to access. Therefore, input–output analysis is used in this study to measure the footprint of the carbon emissions of the forest industry in China based on the Input–Output table (given the research objective and data availability).

### 2.2. Method to Analyze the Factors Affecting Embodied Carbon Emissions

It is equally important to measure the factors driving carbon emissions, based on calculating the total amount of carbon emitted. After measuring the carbon emissions, the factors affecting it are investigated. In the investigation of the factors driving the emission of embodied carbon, the main approaches are Structural Decomposition Analysis (SDA) and Index Decomposition Analysis (IDA). SDA requires a great deal of accuracy in its dataset and large amounts of structural details are needed. Compared with SDA, IDA can use summarized data, of which the total indicator can be easily decomposed and the impact from each factor analyzed to find prevailing trends. As a result, the IDA literature is characterized by greater detail in terms of time periods and countries investigated (Hoekstra & Bergh, 2003 [25]). Therefore, we have opted for the IDA method to analyze the factors affecting carbon emissions in the forest product industries.

By far, the most often used Index Decomposition Analysis methods are the Laspeyre and Disivia Index methods (Ang & Choi, 2020 [26]), but given the non-decomposable residual items in the Laspeyre method, the Disivia method is preferred because the residuals from this method are much smaller than those of the Laspeyre method (Buongiorno et al., 2017 [27]). The Divisia Index method has two forms, i.e., the Average Mean Divisia Index (AMDI) and the Log Mean Divisa Index (LMDI) (Hoekstra & Bergh, 2003 [25]). Because zero values cannot be accepted in the AMDI form, the LMDI has an advantage and is regarded as the more exact form in the IDA approach; both its multiplicative and additive forms have the characteristics of consistency and uniqueness. It provides the perfect decomposition solution without any residual term, is easily explained and allows even zero values. Therefore, we have opted for the LMDI in order to analyze embodied carbon factors' contributions to emissions.

### 3. Measurement of Embodied Carbon
### 3.1. Data and Industries

We analyzed the embodied carbon of the forest products sector in 2002, 2007 and 2012, 2017. The direct consumption coefficient matrix of each industry was obtained from the China Input–Output Tables for these four years. The industries in the China Input-Output Tables (China Input–Output Association, [28–33]) do not correspond with the energy coefficients in the Energy Consumption Tables (National Statistics Bureau, 2018 [34]), since there are 42 industries in the Input–Output Tables, but the Energy Statistics

Yearbook recognizes only 28 industries. Given that the Energy Statistics Yearbooks were the basis from which to obtain accurate energy consumption data for the industries that we studied, we needed to adjust the forest industry in the Input–Output Tables to be line with those identified for the purpose of energy consumption. The adjusted industries are given in Table 1. Our purpose was to calculate the total embodied carbon emissions, and the carbon dioxide coefficients that we used are presented in Table 2.

**Table 1.** Adjusted industry classification.

| Industry Code No. | Industry |
|---|---|
| 01 | Agriculture, forestry, livestock, fishing |
| 02 | Coal mining |
| 03 | Petroleum and natural gas exploitation industry |
| 04 | Metal mining industry |
| 05 | Non-metal mining industry |
| 06 | Food and tobacco processing industry |
| 07 | Textile industry |
| 08 | Textile product industry |
| 09 | Wood furniture manufacturing industry |
| 10 | Paper, stationery manufacturing industry |
| 11 | Fuel processing |
| 12 | Chemical industry |
| 13 | Non-metal product industry |
| 14 | Metal processing industry |
| 15 | Metal product industry |
| 16 | General and special equipment manufacturing |
| 17 | Transportation equipment manufacturing industry |
| 18 | Electrical equipment manufacturing industry |
| 19 | Electronic equipment manufacturing industry |
| 20 | Office machinery manufacturing |
| 21 | Other manufacturing |
| 22 | Electric power supply industry |
| 23 | Production and supply of gas industry |
| 24 | Production and supply of water industry |
| 25 | Construction industry |
| 26 | Transportation, storage and communications |
| 27 | Trade, accommodation and catering industry |
| 28 | Other industries |

Source: China Energy Yearbook 2018.

### 3.2. Model and Computations

We used the input–output method proposed by Leontief (1930) to calculate the embodied carbon in all Chinese industries [35]. This method is a quantitative approach to study the input–output relations in all sectors in an economy and is a powerful tool to study the resources or pollution embodied in commercial goods and services [36]. The approach rests on Leontief's basic assumption of the constancy of the input coefficient of production and the linear structure of the economy. According to the input–output balance, the model, using matrix notation, can be written as follows:

$$X_t = AX_t + Y_t \ (t = 2002, 2007, 2012, 2017) \tag{1}$$

where $a_{ij} = \frac{x_{ij}}{x_j}$ $A = (a_{ij})$ ($i, j = 1, \ldots, 28$ and denote different branches of industries) denotes the direct consumption coefficient, X is the vector of gross input, Y is the vector of social gross output. This equation can be solved for X as follows:

$$X_t = (I - A_t)^{-1} Y_t \tag{2}$$

where $(I - A_t)^{-1}$ is the inverse Leontief matrix, i.e., the complete consumption coefficient matrix.

**Table 2.** $CO_2$ emission coefficients by energy source.

| Code | Energy Source | $CO_2$ Emission by 1 kg Standard Charcoal Equivalent (Unit: kg) | Calorific Value (Unit: kcal/kg) | $CO_2$ Emission Coefficient (Constant) |
|------|---------------|------------------------------------------------------------|----------------------------------|-----------------------------------------|
| 01 | Raw coal | 94,600 | 6800 | 2.69 |
| 02 | Washed clean coal | 94,600 | 6800 | 2.69 |
| 03 | Other washed coal | 94,600 | 6400 | 2.53 |
| 04 | Molded coal | 97,500 | 3800 | 1.55 |
| 05 | Coke | 107,000 | 7000 | 3.14 |
| 06 | Coke oven gas | 44,400 | 5000 | 2.07 |
| 07 | Other oven gas | 44,400 | 4200 | 0.89 |
| 08 | Other coking products | 107,000 | 7000 | 3.69 |
| 09 | Crude oil | 73,300 | 9000 | 3.41 |
| 10 | Petrol | 70,000 | 7500 | 3.05 |
| 11 | Kerosene | 71,900 | 8500 | 3.20 |
| 12 | Diesel oil | 74,100 | 8800 | 3.17 |
| 13 | Fuel oil | 77,400 | 9200 | 3.76 |
| 14 | Liquefied petroleum gas | 63,100 | 6635 | 3.02 |
| 15 | Refinery dry gas | 57,600 | 9000 | 2.41 |
| 16 | Other petroleum products | 73,300 | 9000 | 3.07 |
| 17 | Natural gas | 56,100 | 8900 | 2.09 |

Source: Energy Statistics Yearbook 2018.

Next, we present the steps that are needed to be able to calculate the total amount of embodied carbon.

First, the direct amounts of carbon emission are calculated for the adjusted industries that are presented in Table 1. Let $j$ denote a specific industry, and $\beta_i$ denote the carbon dioxide emission coefficient of energy $i$, and $f_{ji}$ the consumption of energy $i$ from industry $j$. The direct carbon emissions of industry $j$ can then be calculated as in Equation (3).

$$F_j = \sum_{i=1}^{17} f_{ji}\beta_i \tag{3}$$

Because the unit of $CO_2$ emission of the various energy sources is $kgCO_2$ (TJ), while the consumption units of the various kinds of energy are kg, liters and $m^3$, it was necessary to reconcile the $CO_2$ emission coefficients in order to agree with the emission units—i.e., $kgCO_2$ (kg), $kgCO_2$ (L) or $kgCO_2$ ($m^3$) are converted as follows:

$$\beta_i = \beta'_i/10^9 \times \alpha \times K_i \tag{4}$$

where $\beta'_i$ denotes the $CO_2$ emission coefficient before conversion, $\alpha$ is the conversion coefficient of kcal into J with the standard value of 4.1868 (J/Kcal), $K_i$ denotes the heat value of the energy used by industry j in Kcal (kg), $kgCO_2$ (L) or $kgCO_2$ ($m^3$). The results are shown in Table 2.

If we let E denote the direct carbon emission matrix, then the domestic gross output of industry $j$ is $X_j$; therefore, the direct carbon emission intensity matrix $E$ is obtained from Equation (5) as follows:

$$E_j = \frac{F_j}{X_j} \tag{5}$$

The embodied carbon emission intensity matrix $W_t$ can be obtained based on the direct carbon emission matrix. Using $(I - A_t)^{-1}$, the complete consumption coefficient matrix, the intensity matrix is computed as follows:

$$W_t = (I - A_t)^{-1} \times E_t \qquad (6)$$

The domestic embodied carbon emission is comprises two parts, i.e., the carbon emitted by exported products and emissions from domestic products, denoted by the matrices $C_t^e$ and $C_t^d$, respectively. The total carbon emissions are $C_t = C_t^e + C_t^d$, where $C^e$ and $C^d$ can be calculated using (7) and (8) below, respectively, and where $U^e$ is the final use vector of exported products of each industry and $U^d$ the final use vector of each domestic industry.

$$C_t^e = W_t \times U_t^e \qquad (7)$$

$$C_t^d = W_t \times U_t^d \qquad (8)$$

Our model focuses on the carbon emissions from net exports and final domestic use, in view of the emission source that it is based on, by definition excluding imported material such as imported round-wood or lumber.

Using the LMDI model, the energy structure, energy efficiency, economic output and population size are analyzed to be able to obtain their contribution to embodied carbon emissions in the forest product industries. The extended LMDI model equation is written as follows:

$$C = \sum_i \frac{C_i}{E_i} \times \frac{E_i}{E} \times \frac{E}{Y} \times \frac{Y}{P} \times P = \sum_i F_i \times S_i \times I \times G \times P \qquad (9)$$

where $C$ is the direct carbon emission, $i$ the energy type, $C_i$ the direct carbon emission from energy $i$ in the $F_i$, $E$ the energy consumption and $E_i$ the consumption of energy $I$; $Y$ denotes the total output of the forest industries and $P$ is the size of the population. Let us define $F_i = \frac{C_i}{E_i}$ as the carbon emission intensity from energy $i$, i.e., the carbon emission factor. $S_i = \frac{E_i}{E}$ denotes the ratio of energy $i$ in energy consumption and is referred to as the energy structure. $I = \frac{E}{Y}$ denotes the energy consumption per unit of output, i.e., the energy factor, and $G = \frac{Y}{P}$ denotes the per capita output of the $F_i$, referred to as the economic output factor. Let $C^0$ denote the embodied carbon in the base period, $C^t$ the embodied carbon in the reporting period and $\Delta C$ the change in the embodied carbon during the five-year interval between the base and reporting periods. Then, the change in the embodied carbon is obtained as follows:

$$\Delta C = C^t - C^0 = \Delta C_P + \Delta C_S + \Delta C_I + \Delta C_G \qquad (10)$$

The contribution of each factor—for example, the factor $X$—is computed from Equation (11):

$$\Delta C_S = \sum_i \frac{C^t - C^0}{\ln C^t - \ln C^0} \times \ln \frac{X_i^t}{X_i^0} \qquad (11)$$

Given that the carbon dioxide emission coefficient, i.e., the carbon emission factor from each energy, is a constant, the $\Delta C_F$ in (10) remains zero. Hence, Equation (10) can be reduced to Equation (12):

$$\Delta C = C^t - C^0 = \Delta C_S + \Delta C_I + \Delta C_G + \Delta C_P \qquad (12)$$

## 4. Results and Analysis

### 4.1. Embodied Carbon Emission

Using Equations (2)–(8), we calculated the embodied carbon emissions of China in 2002, 2007, 2012 and 2017 to be 2828, 3256, 4021 and 5394 Mt, respectively, i.e., there was a considerable increase in emissions over every five-year interval of our study period. By far,

the largest part of this carbon is emitted from domestic use, while exports accounted for 19.66%, 30.30% and 18.90%, 13.61% of the total emissions, respectively, during 2002–2017 (see Figure 1). Meanwhile, the forest industry embodied carbon emissions increased during the first three 5-year periods, but declined in 2012–2017. The embodied carbon of exported forest industry products was 27.85% of the total carbon of the forest industry products in 2002, 24.57% in 2007, 27.12% in 2012 and 29.84% in 2017 (see Figure 2). The average share of exported carbon in the forest industry in the last decade was higher than the exported proportions of emissions of all industries. The high level of carbon shares in FI is due to the export trade in this sector.

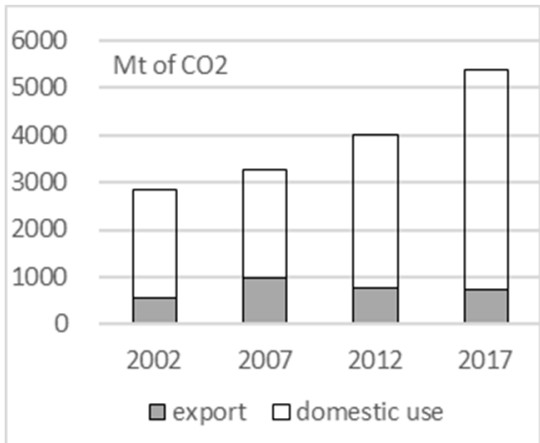

**Figure 1.** Embodied carbon emissions in China.

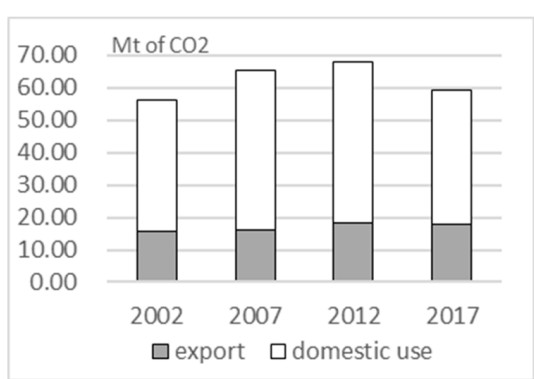

**Figure 2.** Embodied carbon emission in Forest Industry.

The embodied carbon varies between different industries. Transportation, trade and restaurant sectors and the chemical industry fall into the top categories. Pulp and paper and furniture industry ranked 16th and 18th (see Table 3).

**Table 3.** Decomposition of embodied carbon emissions of the forest products industry (Mt).

| Period | Energy Structure $\Delta C_S$ | Energy Intensity $\Delta C_I$ | Economic Output $\Delta C_G$ | Population Size $\Delta C_P$ | Carbon Emission $\Delta C$ |
|---|---|---|---|---|---|
| 2002–2007 | −6.56 | −18.23 | 32.69 | 1.11 | 9.02 |
| 2007–2012 | −5.48 | −24.93 | 31.85 | 1.31 | 2.75 |
| 2012–2017 | −3.42 | 21.32 | −27.26 | 0.65 | −8.7 |
| Accumulative | −15.46 | −21.84 | 37.28 | 3.07 | 3.07 |

Source: Energy Statistics Yearbook 2012.

### 4.2. Embodied Carbon of Forest Product Industries

If we join the furniture and the paper industries (industry codes 9 and 10), the embodied carbon and direct carbon emission for 2002, 2007, 2012 increased over time but decreased from 2012 to 2017 (see Figure 3). The direct carbon increased from 43.21 Mt in 2002 to 69.03 Mt in 2017, while the embodied carbon increased from 33.68 to 45.36 Mt in 2017.

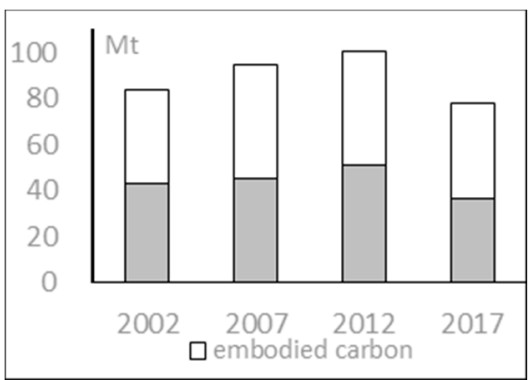

**Figure 3.** Carbon emissions by FI in China.

The embodied carbon intensity of the forest products industry in China, expressed as carbon dioxide emissions per unit output value in thousand RMB (the Chinese currency; 1 USD is 6.5 RMB approximately), has been decreasing (see Figure 4). The direct carbon intensity dropped dramatically from 39.16 to 5.55 t per thousand RMB. The embodied carbon intensity dropped from 150.79 to 21.43 t per thousand RMB. In relative terms, the direct carbon intensity decreased by 85.81% between 2002 and 2017, while the embodied carbon decreased by 85.78% during this period. This suggests that the carbon emission per unit output in the FI is declining and its manufacturing technology has improved during the last decade.

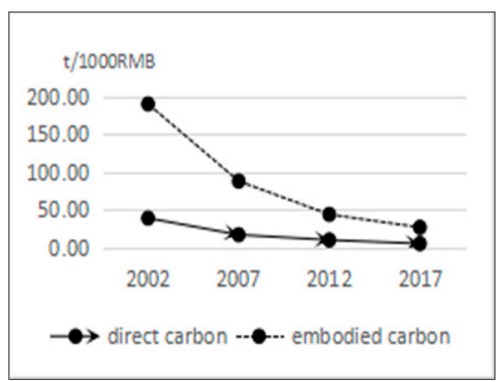

**Figure 4.** Carbon emission intensity of the Forest Industry

### 4.3. Analysis of the Factors Affecting Embodied Carbon

It is of interest to investigate the factors affecting the embodied carbon emissions dynamics in the forest products industry in China during 2012–2017. Using 2002–2007 as the base period, the factors affecting embodied carbon emissions can be decomposed as shown in Table 3. Except for energy intensity and energy structure, economic output and population increase the quantity of embodied carbon. Economic output is the dominating factor, contributing the most to embodied carbon and accounting for 611% of the total change in embodied emissions during the 2002–2017 period. At the same time, energy intensity accounted for −451% of the total change and thus decreased the embodied emissions. The proportional contribution of the population amounts to 18% of the total

increase on average, while the change in the energy structure amounts to −78% of the embodied carbon emissions during the 2007–2017 period.

The decomposition of embodied carbon emissions in Table 3 reveals that the energy intensity drove embodied carbon emissions from negative in 2002–2012 to positive in 2012–2017, while economic output had the opposite effect during the same period. China's forest products industry expanded during the ten-year period of 2002–2012 but shrank in 2012–2017. In contrast, the population continued to increase, also increasing the embodied carbon emissions, though only moderately. The change in energy structure decreased from 2002 to 2017, leading to a continuous decrease in embodied carbon. The contribution of energy structure is to decrease the embodied carbon, which indicates less carbon released during each 5-year period, a sign of improving energy structure and also more clean energy used to decrease the carbon emissions.

## 5. Conclusions and Discussion

The embodied carbon emissions in China's forest product industries increased slowly, with exported carbon accounting for 38.59% of the total carbon in 2002, for 32.57% in 2007, 37.22% in 2012 and 42.54% in 2017. These shares are considerably higher than the average respective shares of all industries. In view of the high levels of carbon emissions in the forestry sector, importing wood products might be a good choice for China and maybe other countries which rely on older production technologies as compared to manufacturing these products domestically. China's export-oriented trade in the forest products industry caused the country to assume greater amounts of embodied carbon emissions than would have been the case if production had been conducted only domestically. The situation is similar to in all industries in China, where China has a large outflow of carbon burden in its exports [3].

The measurement of the embodied carbon in five-year intervals from 2002 to 2017 indicates that the total embodied carbon of the FI in China has been increasing, but the rate of increase slowed down and actually started to decrease in 2017, since the rate of increase was 16.03% in 2007, 4.22% in 2012 and −12.79% in 2017. For the first five years, embodied carbon in the forest product industries increased by 9.02 Mt, but for the second five-year period, the increase was less than half, i.e., 2.75 Mt, and during the third five-year period, embodied carbon in the forest product industries actually decreased considerably. Meanwhile, the increment of total $CO_2$ emission in China shows a steady downward trend and the effects of economic growth are far larger than energy intensity on the increase in China's $CO_2$ emissions [36].

Economic output and energy intensity are the most important and positive driving forces in China's forest product industries. Changing energy structure has decreased the embodied carbon. This indicates that the manufacturers are lowering the production costs by using more efficient means of manufacturing. The energy used in the forest product industries still mainly comes from raw coal, though the share of coal decreased from 69% from 2012 to 58% in 2019 [37]. Coal has the highest carbon emissions among all types of energy, even though energy structure does not seem to exert much effect on the reduction in carbon emission. The growing population drives the embodied carbon emission to slightly increase. In the short term, the direction of the development is not likely to change dramatically, but the increase in the efficiency of production processes in the forest product industries is likely to continue, thus decreasing the embodied carbon emissions of the sector.

The forest product industry sector in China is still facing great pressure related to energy intensity in the long run. The raw-coal-dominated energy structure is still lagging far behind that of the developed countries in terms of carbon emissions. This has led to a high level of carbon emissions, despite the efforts to improve the energy efficiency per unit of output.

In order to lower embodied carbon in the forest industry sector in China, we offer the following suggestions.

One feasible strategy is to increase the export of finished products by substituting primary and intermediate products for finished products. Because the FI in China is still labor-intensive, it is urgent to push this industry to improve the economies of scale, enhance added value, increase competitiveness and modernize the production technology in the forest sector. Encouragement of technical innovation in forest industries production should lead this industrial sector in the direction of technical-intensive and knowledge-intensive processes.

An alternative choice is to increase the share of clean energy in the energy structure. For all industries, the proportions of clean energy use in production in developed countries such as the USA, Japan, Germany and France were 34.0%, 46.2%, 44.6% and 64.0%, respectively, in 2018, while it was only 22.2% in China (data source: BP Statistical Review of World Energy 2019 | 68th edition, UK). Increasing the use of hydro-electric and nuclear energy, as well as renewable energy, would be an effective way to reduce carbon emissions. The use of carbon-neutral energy sources such as biomass from the residues and recycled energy resources in forest industries would promote energy efficiency

**Author Contributions:** Conceptualization, methodology and writing—original draft preparation, L.W.; Calculation: Z.C.; Review and editing, J.K.; Validation, Y.S. All authors have read and agreed to the published version of the manuscript.

**Funding:** This research received funding from China forestry technology innovation new platform operation subsidy (2020132017) and Yunan nature ecology monitoring network program (2020-YN-07).

**Institutional Review Board Statement:** Not applicable.

**Informed Consent Statement:** Not applicable.

**Data Availability Statement:** The data used in the paper: (1) National Accounting Department. National Bureau of Statistics: China Input & Output Table, Beijing, China, the access link is: http://www.stats.gov.cn/ztjc/tjzdgg/trccxh/zlxz/trccb/201701/t20170113_1453448.html (accessed on 21 February 2021). (2) Energy Statistics Yearbook 2018, Beijing China, the accesslink, http://www.stats.gov.cn/tjsj/tjcbw/201909/t20190924_1699094.html (accessed on 21 February 2021).

**Acknowledgments:** We would like to thank the China Scholarship Council (CSC) for this financial support (CSC No. 201906510062).

**Conflicts of Interest:** The authors declare no conflict of interest. The funders had no role in the design of the study; in the collection, analyses, or interpretation of data; in the writing of the manuscript, or in the decision to publish the results.

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
