# Peer review of "Does Forest Industries in China Become Cleaner? A Prospective of Embodied Carbon Emission"

_sustainability, doi:10.3390/su13042306_

Round 1

Reviewer 1 Report

The manuscript includes information that is interesting on its own merits.  However, the Discussion requires improvement.  This section should generalize regarding the larger meaning of the results, relation of the results to those of other, similar studies, and the implications of the results to other fields of study.  Much of what should be in the Discussion is missing. 

Carbon emission intensity is a measure of the efficiency of a process, but total emissions are arguably more relevant in today’s world of changing climate.  This should, at least, be acknowledged.  Also, placing both sets of Figure 4’s  data on the same scale may be a bit misleading.  I estimate from the figure that the rate of decline of both measures is about the same (an exponential decline of ca. 12% per year).

Other specific comments are listed below:

Line

Comment

31

“Forest Industry (panel, furniture, pulp and paper industry) are the sectors with high energy consumption and high carbon emission”  - provide context (i.e., what does “high” mean?)

39, 71 et seq

Paper should be phrased in the past tense (the investigation has already been done)

49-61

The citations demonstrate a research interest in embodied emissions, but what would improve this introduction is a summarization of what the studies have revealed.

59

Delete on

64

“The objective of this research is to investigating the carbon embodied emission dynamics…”  I would call this a goal (IMO objectives are measurable; investigation is not per se).  Also change “to investigating” to “to investigate”

85-86

“It is implemented by integrating various databases as references for users and requires more detailed data set.”  This is not a useful description of LCA!!

93

Factors are investigated

110-111

Delete

Table 1

Perhaps I don’t understand, but since the paper focuses on the forest products industry, is it necessary to include these other, non-studied industries?  Can this be available online?

Table 2

Column headers must provide unambiguous units.  Calorific value should be kcal kg-1 (or kcal/kg).  I am unsure what the CO2 emission coefficient is

131

What do i and j represent?

175

Total embodied C emissions, correct?

176

I am unsure of the journal requirements, but I have always inserted a space between the value and the units.  Additionally, a comma (e.g., 2,828) makes reading easier. 

Over this period the growth of C emissions is exponential.  Worth pointing out that FI emissions are decidedly not?

Figs. 1, 2

Plots should use same shading scheme for domestic, export.

 I also believe graphics should be stand alone, meaning the reader should be able to understand what is presented without returning to the text for interpretation.  That means axes are unequivocally labeled (for example, are carbon emissions in Fig. 1 Mt of carbon, or carbon dioxide?)

194

RMB acronym is undefined

215

The decomposition of embodied carbon (Table 3) is interesting, but needs more background information for those not versed in the method.  Are these values coefficients?  Do they have units?.  Does carbon emission ΔC be summarize the preceding ΔC values? 

218

Are there no other studies against which you can compare your results?

Also, I suggest combining the Conclusions and the Discussion subsections.   

220-221

Sentence requires support (reference)

222

Meaning of “importing wood products is an optimal choice for the countries in demand” is unclear

224-225

If “the total embodied carbon of the FI in China is increasing, but the rate of increase is slowing down” refers to Fig. 2 (or 3), the statement isn’t accurate.  Over the last period, EC declined, making the discussion of trend difficult (as 225-227 states)

230

“more environmentally friendly” is troublesome, since carbon emissions are undoubtedly increasing

231

“mainly in the form of raw coal” – support for this assertion is needed

233

The last sentence is unclear

241-242

“improve economies of scale, enhance added value, increase competitiveness and modernize production technology” - can you provide an example of each

Acceptable citation format?

Author Response

Thank you for the comments. We revised the discussion part to make it more complete and clear. Because most existing literature focused either on main industries or whole industries, we could not find similar studies to compare the result in forestry sector. However we tried to relate the result to the whole economy context, compared the results of this study with the development of total carbon emissions and the driving factors behind total emissions, though other studies mainly focusing on current situation. The 20 years research period is a novel feature of our study and gives the possibility to investigate the dynamic change of forest industry in China.

Reviewer 2 Report

The use of Input-Output methodology is a good approach.  However, the basic assumption of I/O is linear and fixed coefficient production functions.  This should be pointed out in the article.  The requirement of fixed coefficient production functions is alright when one is not deviating from present conditions.  However, with greater changes the reader should be made aware of the limitations of I/O analyses.

Author Response

Thank you very much for  the comments and pointing out assumptions and limitation in I/O approach, we added the assumption in section 3.2 , line128-130.

As  well ,we have edited the language and the format of the article.

Best regards,

Round 2

Reviewer 1 Report

My only remaining comment regards Figure 4.  The absolute decline 2002-2017 is, in fact, greater for embodied C than for direct C.  The relative rate of decline (fraction per year) is about the same.  Although I don't have the numerical data, but I am confident that I estimated accurately the C values.  If the two data sets are plotted on separate scales that encompass the different ranges, the curves will overlap.  I realize now this is a different argument, but still - in my opinion- worth noting.